# The impact of the COVID-19 pandemic on the use of restraint and seclusion interventions in Ontario emergency departments: A population-based study

Meghan Weissflog[1]*, Soyeon Kim[1,2], Natalie Rajack[1], Nathan J. Kolla[1,3,4,5]*

1 Waypoint Research Institute, Waypoint Centre for Mental Health Care, Penetanguishene, Ontario, Canada, 2 Faculty of Health Sciences, McMaster University, Hamilton, Ontario, Canada, 3 Forensic Psychiatry Division, Centre for Addiction and Mental Health, Toronto, Ontario, Canada, 4 Department of Psychiatry, University of Toronto, Toronto, Ontario, Canada, 5 Department of Psychiatry, University of Saskatchewan, Saskatoon, Saskatchewan, Canada

* mweissflog@waypointcentre.ca (MW); nathan.kolla@camh.ca (NJK)

**Data Availability Statement:** Data used in the present study came from the National Ambulatory Care Reporting System (NACRS; https://www.cihi.ca/en/national-ambulatory-care-reporting-system-

## Abstract

While COVID-19 impacted all aspects of health care and patient treatment, particularly for patients with mental health/substance use (MH/SU) concerns, research has suggested a concerning increase in the use of restraint and seclusion (R/S) interventions, although results vary depending on facility type and patient population. Thus, the present study sought to explore COVID-related changes in the use of R/S interventions among patients presenting to Ontario emergency departments (EDs) with MH/SU complaints. To determine whether temporal and clinical factors were associated with changes in R/S use during COVID, binary logistic regression models were computed using data from the National Ambulatory Care Reporting System database. We then compared both prevalence rates and probability of an R/S event occurring during an ED visit in Ontario before and after the onset of COVID. The number of ED visits during which an R/S event occurred for patients presenting with MH/SU concerns increased by 9.5%, while their odds of an R/S event occurring during an ED visit increased by 23% in Ontario after COVID onset. Similarly, R/S event probability increased for patients presenting with MH/SU concerns after COVID onset (0.7% - 21.3% increase), particularly during the first wave, with the greatest increases observed for concerns associated with increased restraint risk pre-COVID. R/S intervention use increased substantially for patients presenting to Ontario EDs with MH/SU concerns during the first wave of COVID when the strain on healthcare system and uncertainty about the virus was arguably greatest. Patients with concerns already associated with increased R/S risk also showed the largest increases in R/S probability, suggesting increased behavioural issues during treatment among this population after COVID onset. These results have the potential to inform existing policies to mitigate risks associated with R/S intervention use during future public health emergencies and in general practice.

metadata-nacrs), which is owned and administered by the Canadian Institute for Health Information, and, as such, cannot be shared publicly under existing data sharing agreements. These data can be obtained by submitting a data request to the Canadian Institute of Health Information (CIHI; https://www.cihi.ca/en/access-data-and-reports/data-holdings/make-a-data-request).

**Funding:** The author(s) received no specific funding for this work.

**Competing interests:** The authors have declared that no competing interests exist.

## Introduction

In line with the global response to the emergence of the novel coronavirus 2019 (COVID-19), the Ontario Provincial Government declared COVID to be a Public Health Emergency on March 17, 2020 [1]. Predictably, the implementation of a number of public health measures to reduce the spread of infection greatly impacted the Ontario healthcare system [2–6], which, in turn, greatly impacted patient treatment options and experiences [7–9]. Emergency departments (EDs), in particular, saw a substantial decrease in the number of patients accessing emergency services, largely due to public health restrictions and fears of COVID exposure in hospital settings [4, 10–12]. This hesitancy to access emergency services had a disproportionate effect on already vulnerable and underserved patient populations during the pandemic, namely those suffering from mental health and/or substance use (MH/SU)-related disorders, for whom emergency services were a primary source of assistance for some prior to the COVID pandemic [13–15]. Of particular concern, research into the impact of COVID on emergency service provision has shown a significant reduction in both psychiatric care and substance use treatment resources during the pandemic [9, 16–18] despite the notable increase in MH/SU-related concerns during this time [19–26].

In light of the observed reduction in both emergency and psychiatric service access due to the COVID pandemic, one aspect of patient treatment that deserves further exploration is whether the pandemic also impacted the use of restraint and seclusion (R/S) interventions amongst patients during service provision in emergency care facilities. The impact of COVID on the use of R/S interventions during treatment is of concern for a number of reasons, including the fact that COVID was an unprecedented event that affected service provision at every level of the healthcare system [2–6]. Given the fact that the frequency of R/S intervention use is considered an indicator of the quality of care [27–30], due to the very real potential for physical (e.g., asphyxiation, fractures, muscle atrophy and contracture, etc.) and psychological injury (e.g., agitation, delirium, PTSD, etc.) [31, 32], understanding how R/S use was impacted by the pandemic offers a useful measure of the broader impact of COVID on health care system and patient experiences.

Existing research on COVID-related changes in R/S intervention use is in its early stages, with reported changes in R/S use varying substantially depending on the patient population and hospital setting. For example, a study from a tertiary mental health hospital in Ontario, Canada [33] noted a substantial reduction (49–100%) in the use of mechanical restraints in adolescent, forensic, and geriatric patient programs after the implementation of COVID-related restrictions. However, they only sampled data for the first 6 weeks after the COVID onset. Similarly, a high-needs psychiatric ward in Dublin, Ireland [34] reported substantial reductions in the use of R/S interventions after voluntarily reducing ward occupancy to facilitate social distancing and other infection prevention strategies. Specifically, reducing bed availability by only 5% resulted in the number of R/S events decreasing by more than 50% over nine months following the onset of the COVID pandemic. On the other hand, research from a mental health hospital in Malaga, Spain, found no change in the likelihood of mechanical restraint [35].

Most consistently, though, researchers have reported increased use of R/S interventions after the onset of the COVID pandemic. For example, a substantial increase in the incidence rate of physical restraint use was observed in geriatric dementia patients presenting to an acute care hospital in Japan, reaching a 50% likelihood of a restraint event approximately four months after the enactment of nationwide public health measures [36]. Similarly, orders for physical restraint interventions among older adults during acute care hospitalization in Ontario, Canada, were found to have increased in prevalence by 2.4% [37], while children and

youth presenting with mental health concerns to a pediatric hospital in Michigan, USA were found to be almost four times more likely to experience a restraint event after the onset of the COVID pandemic [38]. A retrospective study on youth and adolescent psychiatric hospitalizations in Israel also noted a small increase in R/S intervention use despite a significant decrease in the number of admissions after the onset of COVID [39]. Similarly, a study of adult psychiatric admissions in Germany found that the number R/S events increased by almost 25% for patients who were voluntarily admitted, and 13% among those who were involuntarily admitted during the pandemic [40]. Finally, a study from the Department of Psychiatry at the Geneva University Hospital in Switzerland reported a slight increase in the number of seclusion events after COVID onset, despite a decrease in both voluntary and involuntary hospital admissions [41].

## Current study

With few exceptions, the emerging research described above suggests that the use of R/S interventions increased after the onset of the COVID pandemic, although the degree to which R/S use changed appears to depend on a number of factors, including facility type, patient population, and the institution of COVID-related public health policies and restrictions. However, it is currently unknown whether R/S intervention use changed in other clinical settings, such as EDs, which saw dramatic changes in service provision during the pandemic [4, 10–12]. Similarly, existing research has focused solely on R/S use for patients with MH issues [33–41], despite the fact that SU-related concerns increased on par with MH-related concerns during the pandemic [19–26] and are a known risk factor for the use of R/S interventions [42, 43].

As such, the purpose of the present study was to explore whether specific temporal and clinical case factors were associated with changes in the likelihood of experiencing an R/S event among patients presenting to Ontario EDs with MH/SU-related complaints after the onset of the COVID pandemic. In light of the fact that the frequency of R/S events is a useful indicator of both quality of care and patient experience [27–30], understanding how the onset of the pandemic affected the use of R/S interventions offers important insight into the broader impact of COVID among MH/SU patients seeking emergency care. To this end, the current study employed data from the National Ambulatory Care Reporting System (NACRS) [44] to compare both the prevalence rate and probability of an R/S event occurring during an ED visit before and after the onset of COVID to determine if specific temporal and clinical factors were associated with changes in the probability of using R/S interventions.

## Materials and methods

### Sample

Data for this study were obtained from the NACRS database of emergency care service provision across Canada on November 10, 2021. The raw dataset consisted of administrative and clinical data from 559,817 separate ED visits in Ontario during the fiscal years 2019 (N = 298,734) and 2020 (N = 261,083). These data are anonymized by CIHI prior to dissemination and do not contain information by which an individual accessing ED services could be personally identified.

Due to the nature of the present study, we limited our sample to cases with complete data regarding R/S intervention use. The availability of this data is subject to the data collection instructions set out by NACRS [45], which mandate reporting of R/S intervention use data only for ED patients presenting with a main problem diagnosis (see Clinical Variables below) from the Mental and Behavioural Disorders (Chapter V) or Intentional Self-Harm categories of the ICD-10-CA [46, 47], regardless of whether an R/S even occurred. Thus, to ensure

complete R/S use data, we restricted our sample to cases in which an individual presented to an Ontario ED with an MH/SU-related primary concern for the present analyses, resulting in a final sample of data for 332,046 individual ED visits (2019: $N = 174,686$ cases; 2020: $N = 157,360$ cases; Table 1).

## Study measurements

**Occurrence of R/S intervention event.** The use of any R/S intervention is documented in the NACRS database by the type and duration of the intervention that was used. We derived a binary variable indicating whether or not an R/S event had occurred during each recorded visit in the dataset by taking the type of R/S intervention used and recoding all R/S intervention types as "1", indicating that an R/S event had occurred during that visit, while all instances of "none" were recoded to "0", indicating that no R/S event occurred.

**Type of R/S intervention event.** We also derived a categorical variable to indicate which type of R/S intervention occurred during each recorded visit in the dataset. We took the type

**Table 1. Sample descriptives for clinical and temporal variables of interest for ED visits in Ontario during the 2019 and 2020 fiscal years.**

| | 2019 (N = 174,686) | | | 2020 (N = 157,360) | | | Change (2020–2019) | | |
|---|---|---|---|---|---|---|---|---|---|
| | *N* | *%* | *P(R/S)* | *N* | *%* | *P(R/S)* | *N* | *%* | *P(R/S)* |
| **R/S Event Occurred** | | | | | | | | | |
| No | 165,894 | 95.0% | — | 147,735 | 93.9% | — | -18,159 | -10.9% | — |
| Yes | 8,792 | 5.0% | **5.0%** | 9,625 | 6.1% | **6.1%** | 833 | 9.5% | *1.1%* |
| **Type of R/S Event** | | | | | | | | | |
| Restraint | 6,616 | 3.8% | **3.8%** | 7,251 | 4.7% | **4.7%** | 635 | 9.6% | *0.8%* |
| Seclusion | 2,176 | 1.3% | **1.3%** | 2,374 | 1.6% | **1.6%** | 198 | 9.1% | *0.3%* |
| **Month of Visit** | | | | | | | | | |
| April | 14,933 | 8.5% | **4.7%** | 9,436 | 6.0% | **8.3%** | -5,497 | -36.8% | *3.6%* |
| May | 15,797 | 9.0% | **4.9%** | 12,103 | 7.7% | **7.3%** | -3,694 | -23.4% | *2.5%* |
| June | 14,652 | 8.4% | **5.3%** | 13,274 | 8.4% | **6.7%** | -1,378 | -9.4% | *1.4%* |
| July | 14,782 | 8.5% | **5.1%** | 14,651 | 9.3% | **5.9%** | -131 | -0.9% | *0.8%* |
| August | 14,952 | 8.6% | **5.6%** | 14,201 | 9.0% | **5.9%** | -751 | -5.0% | *0.3%* |
| September | 15,074 | 8.6% | **4.8%** | 13,531 | 8.6% | **5.3%** | -1,543 | -10.2% | *0.4%* |
| October | 15,059 | 8.6% | **4.8%** | 13,603 | 8.6% | **5.6%** | -1,456 | -9.7% | *0.8%* |
| November | 14,633 | 8.4% | **4.6%** | 13,354 | 8.5% | **5.7%** | -1,279 | -8.7% | *1.0%* |
| December | 14,249 | 8.2% | **5.2%** | 12,735 | 8.1% | **6.2%** | -1,514 | -10.6% | *1.0%* |
| January | 14,444 | 8.3% | **5.1%** | 13,242 | 8.4% | **6.1%** | -1,202 | -8.3% | *1.0%* |
| February | 13,862 | 7.9% | **4.7%** | 12,337 | 7.8% | **5.8%** | -1,525 | -11.0% | *1.0%* |
| **Main Problem Diagnosis** | | | | | | | | | |
| Alcohol SUD | 35,175 | 20.1% | **4.2%** | 28,966 | 18.4% | **5.0%** | -6,209 | -17.7% | *0.7%* |
| Opioid SUD | 4,393 | 2.5% | **4.1%** | 4,687 | 3.0% | **9.6%** | 294 | 6.7% | *5.4%* |
| CBD SUD | 3,438 | 2.0% | **4.9%** | 3,729 | 2.4% | **15.7%** | 291 | 8.5% | *10.8%* |
| Poly SUD | 15,990 | 9.2% | **9.8%** | 14,786 | 9.4% | **27.6%** | -1,204 | -7.5% | *17.7%* |
| Dementia | 681 | 0.4% | **11.9%** | 756 | 0.5% | **33.2%** | 75 | 11.0% | *21.3%* |
| Schizophrenia | 18,696 | 10.7% | **12.3%** | 20,745 | 13.2% | **30.0%** | 2,049 | 11.0% | *17.7%* |
| Mood Disorder | 29,146 | 16.7% | **4.4%** | 23,598 | 15.0% | **13.9%** | -5,548 | -19.0% | *9.4%* |
| Anxiety Disorder | 58,874 | 33.7% | **2.0%** | 51,955 | 33.0% | **5.4%** | -6,919 | -11.8% | *3.4%* |
| Personality Disorder | 4,133 | 2.4% | **7.1%** | 4,346 | 2.8% | **20.5%** | 213 | 5.2% | *13.3%* |
| Other MH Disorder | 4,160 | 2.4% | **5.8%** | 3,792 | 2.4% | **15.9%** | -368 | -8.8% | *10.1%* |

N = number of visits during fiscal year; % = percentage of visits during fiscal year; P(R/S) = probability of a restraint or seclusion event occurring during a single visit

of intervention and recoded all R/S interventions ((1) No R/S intervention, (2) physical restraint, (3) mechanical restraint, (4) chair restraint, or (5) seclusion room) into one of three categories ("0" = no R/S intervention, "1" = restraint, "2" = seclusion).

**Registration year.**   The data used in the present analyses consisted of client visit data collected at the time of service provision at ED rooms across Ontario, Canada, during either the (1) 2019 and (2) 2020 fiscal year (April 1 –March 31). We chose to focus on this time range because it naturally overlapped with the declaration of COVID as a public health emergency in Ontario in March 2020.

**Registration month.**   The registration month was coded from the date on which the patient officially registered for service provision in an Ontario ED, resulting in twelve categories, one for each month of the year. Because the present data are split based on fiscal and not calendar year, an anomaly exists in the sense that March "2019" was, in actuality, March 2020, the beginning of the COVID public health emergency in Ontario. Thus, we excluded the month of March from later interpretations to avoid an inaccurate representation of the time prior to COVID onset.

**Main Problem Diagnosis (MPDx).**   Patients presenting at EDs in Ontario are assigned an ICD-10-CA code that best describes the most clinically significant reason for the visit (e.g., requiring the greatest use of resources), as judged by the attending physician. As noted above, due to the reporting criteria for R/S interventions in Ontario EDs we limited our sample to patients with an MPDx of MH/SU (ICD-10 Mental Health and Behavioural Disorders), divided into ten separate diagnostic categories: (1) alcohol abuse (F100-F109), (2) opioid abuse (F110-F119), (3) cannabinoid abuse (F120-129), (4) multiple drug abuse (F130-199), (5) dementia (F010-F019; F03), (6) psychotic disorders (F20-F29), (7) mood disorders (F30-F39), (8) neurotic (anxiety) disorders (F40-F48), (9) other mental health disorders (F50-F59), (10) personality disorders (F60-69).

## Data analyses

To examine COVID-related changes in R/S in Ontario EDs among patients with MH/SU presentations, we calculated percent change scores between cases before (2019) and after (2020) COVID onset for the number of ED visits, the occurrence of an R/S event and the type of R/S event per visit. In addition, an unadjusted binary logistic regression analysis was conducted predicting the occurrence of an R/S event (yes/no) from the registration year (2019/2020) to determine whether any observed changes in R/S prevalence corresponded to a significant change in the odds of experiencing an R/S event after COVID onset.

We then examined the association of temporal (registration month) and administrative factors (MH/SU MPDx) with changes in R/S intervention use after COVID onset by conducting a series of unadjusted binary regression analyses predicting R/S event occurrence (yes/no) from registration year (2019/2020), entering each factor as a moderator of this relationship (S1 Table). From the resulting regression models, we derived the probability of experiencing an R/S event for each category of the predictor variables in both 2019 and 2020 and calculated the percentage change in probability before and after the COVID onset (S2 Table). We chose to present probability statistics in the present study, as opposed to the more traditional odds ratio (OR) statistics associated with binary logistic regression analyses, due to the fact that ORs, by definition, are derived based on a reference group. In the case of the present data, there is no natural control or reference group/category, making any relative comparisons between predictor categories arbitrary and the results difficult to generalize. All analyses were conducted using SPSS (v.28) [48] and Microsoft Excel (2016) [49].

## Results

### COVID-related changes in ED visits and the use of R/S interventions

Before exploring factors associated with COVID-related changes in R/S intervention use, we first wanted to determine that there were, in fact, significant changes in the number of ED visits and/or the number of R/S events that occurred in Ontario EDs after the onset of COVID. The overall number of visits to Ontario EDs for MH/SU presentations decreased by 9.9% (17,326 visits) after the onset of the pandemic (Table 1 and Fig 1).

While ED visits decreased in general, the number of visits in which R/S interventions were used increased by 9.5% (833 instances; $\chi^2(1) = 185.52$, $p < .001$) during the first year of COVID (Table 1 and Fig 2). This is further confirmed by binary logistic regression results indicating that patients presenting to Ontario EDs during the first year of COVID were 23% more likely to experience an R/S event than those presenting the year prior ($OR = 1.23$, $p < .001$, 95% CI [1.19, 1.27]). This observed increase in the prevalence of R/S intervention use was comparable across both restraint (physical/mechanical/chair; 9.6% increase) and seclusion (9.1% increase) intervention methods (Table 1 and Fig 3).

### Factors associated with COVID-related changes in the use of R/S interventions

**Registration month.** There was a significant interaction between the month and the year of registration ($Wald\ \chi^2\ (11) = 109.48$, $p < .001$), suggesting that the onset of the COVID pandemic had a substantially different effect on the likelihood of an R/S event occurring depending on the month in which a patient visited an ED. Upon exploration of predictor-level interaction effects, a clear pattern of changes in R/S probability emerged primarily centered on the onset of the COVID pandemic in Ontario (Table 1 and Fig 4).

Most notably, there was a dramatic increase in the probability of an R/S event at the start of the first wave of the COVID pandemic in Ontario, beginning with a 3.6% increase in April

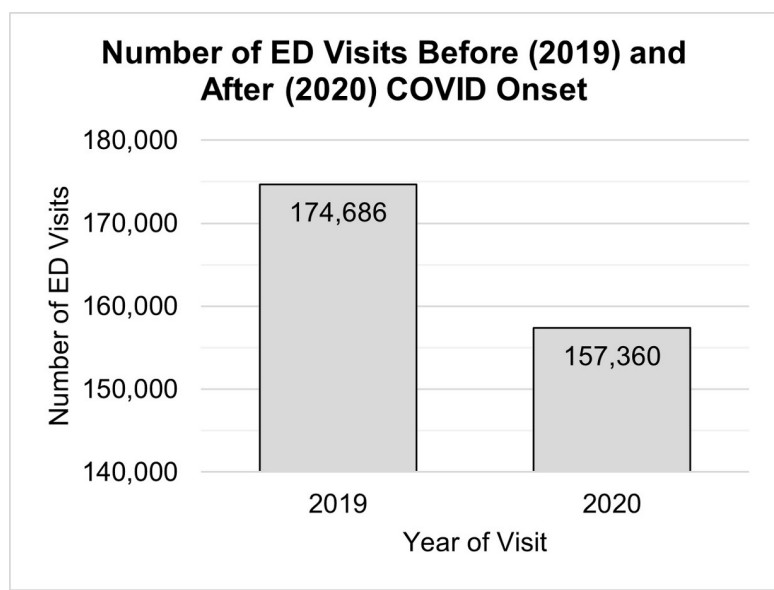

**Fig 1. Number of ED visits by registration year.** Number of ED visits in Ontario for MH/SU concerns before (2019) and after (2020) the onset of the COVID-19 pandemic. ED = emergency department, MH/SU = mental health/ substance use.

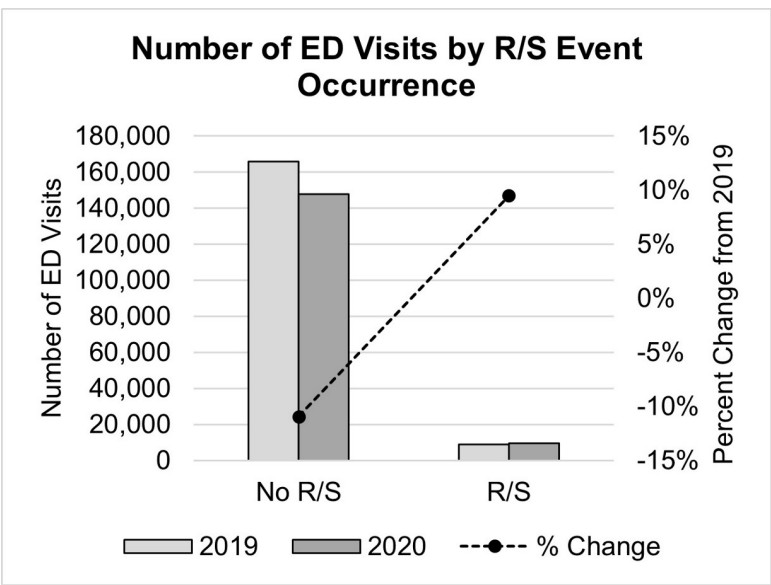

**Fig 2. Number of ED visits by R/S event occurrence.** Number of ED visits in Ontario before (2019) and after (2020) the onset of the COVID-19 pandemic by R/S event occurrence (bar; right axis) with corresponding percent change from pre-COVID (2019) levels (line; left axis). ED = emergency department, R/S = restraint and seclusion intervention.

2020, followed by a decreasing trend in the magnitude of the R/S probability change in May (2.5%), and June (1.4%) of 2020, with R/S probability returning to levels comparable to those before the onset of COVID by July of 2020 (> 1% increase in R/S probability; Table 1 and Fig 4). Another period of increased likelihood of R/S was observed during Ontario's second wave

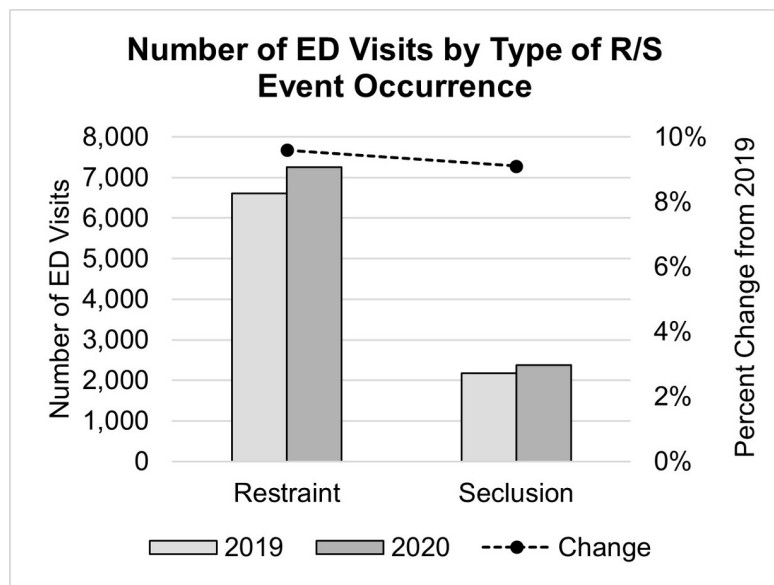

**Fig 3. Number of ED visits by type of R/S event occurrence.** Number of ED visits in Ontario before (2019) and after (2020) the onset of the COVID-19 pandemic by type of R/S event occurred (bar; right axis) with corresponding percent change from pre-COVID (2019) levels (line; left axis). ED = emergency department, R/S = restraint and seclusion intervention.

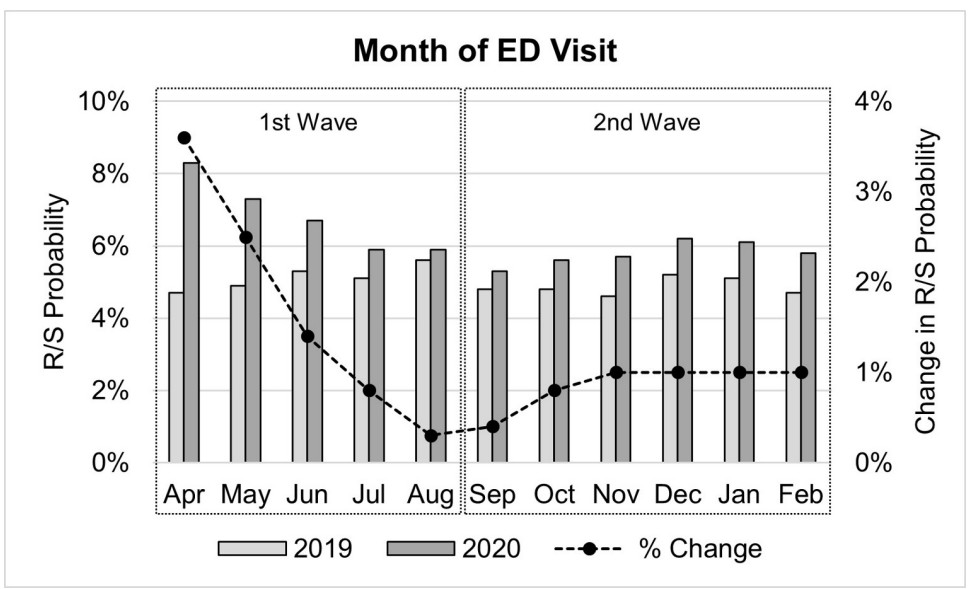

**Fig 4. Month of ED visit.** Percent of ED visits by registration month in which an R/S event occurred before (2019) and after (2020) the onset of the COVID-19 pandemic (bar; right axis) with corresponding changes R/S probability between 2019 and 2020 (line; left axis). ED = emergency department, R/S = restraint and seclusion intervention.

of the COVID pandemic, albeit to a much lesser degree than was seen during the first wave. Specifically, the likelihood of an R/S event began to increase again beginning in September 2020 (the beginning of the second wave) and plateaued at a 1% increase from pre-COVID levels from November 2020 to February 2021 (Table 1 and Fig 4).

**Main problem diagnosis.** The type of SU/MH concern (main problem diagnosis) a patient presented with significantly interacted with patient registration year (*Wald* $\chi^2$ (9) = 30.35, *p* < .001), suggesting that the onset of the COVID pandemic impacted the probability of experiencing an R/S event differently depending on a patient's main problem diagnosis. Further exploration revealed that there was a general increase in the probability of an R/S event across all diagnostic categories from 2019 to 2020 (range = 0.7% - 21.3%), and it is only the magnitude of this increase that varied across diagnostic categories (Table 1 and Fig 5).

Patients with a diagnosis of dementia saw the greatest increase of 21.3% in the likelihood of experiencing an R/S event after COVID onset, followed by patients diagnosed with a psychotic disorder or polysubstance abuse, who both showed an increase of 17.7% in their likelihood of experiencing an R/S event (Table 1 and Fig 5). Individuals diagnosed with a personality disorder saw a moderate 13.3% increase in R/S probability after COVID onset, while an increase of approximately 10% in R/S probability was found for patients diagnosed with cannabinoid abuse, a mood disorder, or other mental health disorder not otherwise specified. Finally, patients diagnosed with alcohol abuse (0.7% increase), opioid abuse (5.4% increase), or an anxiety disorder (3.4% increase) saw the smallest increases in the probability of experiencing an R/S event from pre-COVID levels.

## Discussion

The present study aimed to examine the association between temporal and clinical factors and changes in the use of R/S interventions in Ontario EDs during the COVID pandemic as an indicator of broader changes in the quality of care and patient experiences during this time. To this end, we first wanted to establish whether the onset of COVID affected ED access and the

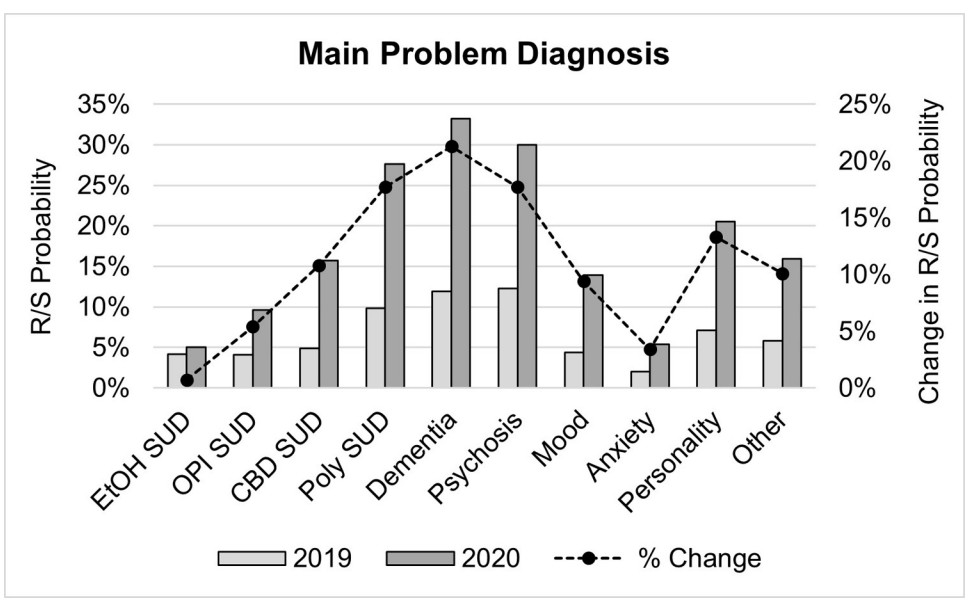

**Fig 5. Main problem diagnosis.** Percent of ED visits by main problem diagnosis in which an R/S event occurred before (2019) and after (2020) the onset of the COVID-19 pandemic (bar; right axis) with corresponding changes R/S probability between 2019 and 2020 (line; left axis). ED = emergency department, R/S = restraint and seclusion intervention.

use of R/S interventions during service provision in Ontario EDs. We found that there were approximately 17,000 fewer visits to Ontario EDs for MH/SU concerns after the onset of the pandemic, a decrease of almost 10% from pre-COVID levels. Notably, despite there being fewer ED visits during the first year of the pandemic, the number of visits where an R/S event occurred increased by almost 10%, which translated to 833 more visits in which the patient was restrained or secluded during service provision, which is consistent with previous research exploring COVID-related changes in R/S intervention use.

Further underscoring the fact that this observed increase in R/S intervention use was related to the pandemic onset, the greatest increase in R/S risk was seen for patients who accessed Ontario EDs during the first few months of the first wave of the pandemic (April–June 2020). The pandemic was officially declared a public health emergency in Ontario on March 17, 2020, suggesting that the observed spike in the probability of R/S intervention use aligns with this declaration and the resulting impact on the public and healthcare system. We suggest that the combination of staffing resource shortages, and strict infection prevention protocols likely prevented staff from effectively employing traditional de-escalation methods and less coercive R/S alternatives, such as 1-to-1 patient supervision. Consistent with this interpretation, we also found that the prevalence of R/S use returned to comparable levels by the end of the first wave of infection, suggesting that, as resources and staffing stabilized, the healthcare system was able to adapt and mitigate factors associated with the temporary increase in the use of R/S interventions.

Even more dramatic than the observed temporal effects of COVID was the finding that the risk of experiencing an R/S event doubled for almost all patients who presented to Ontario EDs with primary complaints associated with MH/SU concerns after the onset of COVID, regardless of the specific disorder. This further supports the suggestion that increases in R/S intervention use during COVID were a function not only of the exacerbation of psychiatric and behavioural symptoms specific to certain disorders but also the systemic issues related to

health care service provision described above, namely, the availability of relevant resources. In addition to increasing the likelihood that patients with MH/SU-related concerns had to turn to ED services during COVID, the lack of alternative resources may have also led patients to delay seeking treatment, resulting in significantly greater symptom severity and in turn, R/S risk when the patient did seek care. Notably, the probability of R/S use nearly tripled for patients presenting with MH/SU concerns that were already associated with increased R/S risk prior to COVID onset, namely psychosis, personality disorder, dementia, and substance abuse. Arguably, these patients are also more likely to have had difficulty complying with strict infection prevention protocols during service provision due to the cognitive and behavioural symptoms specific to these disorders. This in turn may have contributed to the disproportionate increase in R/S intervention use we observed, as the use of physical restraint or patient seclusion may have been necessary to minimize infection risk in non-compliant patients.

## Limitations

The most notable limitations of the present study come from the nature of the sample data used to conduct our analyses. As our present analyses were restricted to data from ED visits in which R/S intervention use reporting was mandatory and complete, this limited our sample to patients who presented with primary complaints associated with an MH/SU condition. In other words, if a patient did not present with an MH/SU, they were not captured in our dataset. While this prevents us from drawing broader conclusions regarding COVID-related changes in R/S intervention use for patients with other primary concerns, it is also a realistic "snapshot" of the use of R/S interventions in an ED setting. Specifically, this population is generally at greater risk of experiencing restraint and needing emergency care, making them an important (and logical) patient population to focus on when trying to understand the impact of COVID on the use of R/S interventions. Similarly, because individual cases in the present data set represent the information collected during a single *visit* to an Ontario ED, the present sample is, by definition, not independent at the subject level in the traditional sense. In other words, there are undoubtedly patients in our sample who presented to an Ontario ED multiple times throughout the fiscal year, resulting in an overrepresentation of certain patient demographics in the sense of the "average" patient accessing ED services.

Finally, the scope of the present study was restricted by the fact that the available data was necessarily limited to the variables collected by the data reporting agency. For example, while it would be informative, the NACRS data collection guidleines do not currently require mandatory reporting of the specific clinical symptoms reported at the time of presentation to the ED or the specific reason for restraint use (e.g., aggressive behaviour, fall risk, treatment compliance, etc.). As such, we cannot draw any conclusions about whether the COVID-related increase in restraint use was associated with an increase of specific behavioural concerns, which may be more prominent among certain patient populations.

## Future directions

This study is just an initial step toward understanding the underlying issues that contributed to the increased use of R/S interventions among patients presenting to EDs with MH/SU-related concerns. Naturally, more research into the factors associated with changes in R/S intervention use due to the COVID pandemic is needed to fully understand how such events impact patient care. A better understanding of the underlying factors driving the differences in the likelihood of an R/S event across different patient populations and treatment settings is of particular importance, not only to identify additional exacerbating factors, but also mitigating factors that effectively reduced R/S risk under similar circumstances. In line with this, the

synthesis of the research into the factors associated with increased R/S intervention use is equally important, as it has the potential to be translated into valuable improvements to existing policies and practices related to the use of R/S interventions. Similarly, continued attention to education and training regarding the use of R/S interventions is equally important, with particular emphasis on appropriate use and practical alternatives to coercive methods. For example, in response to the COVID pandemic, there is a clear need to develop strategies to minimize the need to use such methods as a form of infection prevention among vulnerable populations that struggle with treatment compliance.

In particular, as suggested above, the observed increase in R/S intervention use in the present study is likely not only due to an increase in psychiatric symptom severity but also because of the overall strain on the healthcare systems leading to a lack of resources. Therefore, efforts to prevent similar increases in the number of R/S events during future major public health crises should focus on non-patient-related contextual factors that are likely to be impacted, such as access to non-emergent psychological services and substance abuse programming. One of the outcomes of the pandemic was the advent and refinement of telehealth programs that made non-emergent healthcare services much more accessible while minimizing infection risk. This now-existing infrastructure may help mitigate similar situations during future public health crises by allowing easier and earlier access to services specifically intended to address concerns related to MH/SU issues before they require emergency care and the use of R/S intervention methods. Having said this, the public health system cannot hope to rely solely on telehealth alternatives and should develop appropriate emergency response plans to deploy and allocate resources with the knowledge that major public health crises can have a disproportionately negative impact on the experience of already vulnerable patient populations, as well as patients who may be experiencing MH/SU issue for the first time due to the unique stressors associated with major public health crises, such as COVID.

## Conclusions

The results of the present study provide evidence that the use of R/S interventions in Ontario EDs increased substantially among patients presenting with MH/SU concerns after the onset of COVID, despite a decrease in the number of ED visits during the same period. Further underscoring the detrimental effect of COVID on the Ontario health care system, the observed increase in R/S intervention use was most prominent during the first few months after the public health emergency declaration in Ontario, when resources were under the most significant strain and uncertainty surrounding the virus was at its highest. Finally, we demonstrated that patients presenting to Ontario EDs with MH/SU-related concerns were at a substantially increased risk of experiencing an R/S event during the first year of the pandemic, regardless of the specific nature of their presenting concern, but that this effect was most pronounced for patients with MH/SU concerns traditionally associated with increased risk of R/S intervention use.

Taken together, the findings of the present strongly support the conclusion that the observed increase in R/S intervention use in Ontario EDs may be a result of the general state of upheaval in the Ontario healthcare system caused by the COVID pandemic. In context with the existing literature, the results of this study have the potential to further our understanding of COVID-related changes in the presentation of MH/SU issues in ED facilities, as well as how the healthcare system responded to these changes during the pandemic. Furthermore, this knowledge has the potential to inform and guide improvements to existing policies and procedures related to the use of R/S interventions to minimize the inherent risk to both patients and staff.

## Supporting information

**S1 Table. Binary logistic regression model statistics.**
(PDF)

**S2 Table. Regression equation coefficients (B-weights) and variable values used to calculate R/S probability.**
(PDF)

## Author Contributions

**Conceptualization:** Meghan Weissflog, Nathan J. Kolla.

**Data curation:** Meghan Weissflog, Natalie Rajack.

**Formal analysis:** Meghan Weissflog.

**Methodology:** Meghan Weissflog, Soyeon Kim, Nathan J. Kolla.

**Project administration:** Nathan J. Kolla.

**Supervision:** Nathan J. Kolla.

**Visualization:** Meghan Weissflog.

**Writing – original draft:** Meghan Weissflog.

**Writing – review & editing:** Meghan Weissflog, Soyeon Kim, Natalie Rajack, Nathan J. Kolla.

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
