## [Decision Letter · Decision Letter 0]

6 Nov 2023

PONE-D-23-28304The impact of the COVID-19 pandemic on the use of restraint and seclusion interventions in Ontario emergency departments: A population-based studyPLOS ONE

Dear Dr. Weissflog,

Thank you for submitting your manuscript to PLOS ONE. After careful consideration, we feel that it has merit but does not fully meet PLOS ONE’s publication criteria as it currently stands. Therefore, we invite you to submit a revised version of the manuscript that addresses the points raised during the review process.

We look forward to receiving your revised manuscript.

Kind regards,

Anshuman Mishra, PhD

Academic Editor

PLOS ONE

Journal Requirements:

2. Please note that in order to use the direct billing option the corresponding author must be affiliated with the chosen institute. Please either amend your manuscript to change the affiliation or corresponding author, or email us at plosone@plos.org with a request to remove this option.

Additional Editor Comments:

Article- The impact of the COVID-19 pandemic on the use of restraint and seclusion

interventions in Ontario emergency departments by Meghan et al., 2023 explore COVID-related changes in the use of R/S

interventions among patients presenting to Ontario emergency departments (EDs) with

MH/SU complaints.

These results have the potential to inform existing

policies to mitigate risks associated with R/S intervention use during future public

health emergencies and in general practice.

Article is interesting however required few additional explanations-

1. Clear objective, and conclusion about interventions is help to understand the article more about study.

2. Which kind of clinical feature present in MH/SU complaints.

3. What is impact of Clinical segregation of subjects on the results.

4. Graphical representation is important for better representation.

5. What is future prospect of the study.

6. Any comparative study in same region is important to understand the results.

7. What is public health model for other disease situations (epidemic-sporadic etc.)

Reviewers' comments:

Reviewer's Responses to Questions

**Comments to the Author**

1. Is the manuscript technically sound, and do the data support the conclusions?

Reviewer #1: Yes

Reviewer #2: Yes

2. Has the statistical analysis been performed appropriately and rigorously? 

Reviewer #1: Yes

Reviewer #2: Yes

3. Have the authors made all data underlying the findings in their manuscript fully available?

Reviewer #1: Yes

Reviewer #2: Yes

4. Is the manuscript presented in an intelligible fashion and written in standard English?

Reviewer #1: Yes

Reviewer #2: Yes

5. Review Comments to the Author

Reviewer #1: This is an interesting manuscript in which authors have looked into the impact of the COVID-19 pandemic on the use of restraint and seclusion interventions in emergency departments. In particular authors have looked into COVID related changes in the use of R/S interventions in Emergency departments with mental health/Substance use complaints. The study has good sample size, results and conclusions are aligned with the central message. The study adds to the existing knowledge on R/S use during pandemic.

Reviewer #2: Overall, this study has the potential to further the understanding of COVID-related changes in the presentation of MH/SU issues in ED facilities, as well as the ways in which the healthcare system responded to these changes during the pandemic and the data from this study can be used to improve the existing policies and procedures related to the use of R/S interventions to minimize the inherent risk to both patients and staff.

6. PLOS authors have the option to publish the peer review history of their article (what does this mean?). If published, this will include your full peer review and any attached files.

Reviewer #1: No

Reviewer #2: **Yes: **Amit Kumar Mishra

---

## [Author Response · Author response to Decision Letter 0]

21 Feb 2024

Please find below our responses to the reviewers' comments on our manuscript titled “The impact of the COVID-19 pandemic on the use of restraint and seclusion interventions in Ontario emergency departments: A population-based study” (PONE-D-23-28304), which was submitted to PLOS ONE in September of 2023. I would like to express my gratitude to the reviewers for their constructive feedback, which has been helpful in revising our manuscript. Please find below detailed responses to each of the points raised by the reviewers.

1. Clear objective, and conclusion about interventions is help to understand the article more about study.

The reviewers correctly highlighted the importance of a clear objective and conclusion. In response to this, we have revised the manuscript to provide a more focused objective by elaborating on the fact that COVID-related changes in the use of restraint and seclusion interventions can be viewed as an indicator of the broader impact the COVID-19 pandemic had on patient experiences when seeking emergency care [pg. 5, lines 69-76; pg. 7, lines 119-123]. Additionally, we have added a “Future Directions” section to our conclusions to elaborate on avenues for potential intervention that may aid in preventing similar situations during future healthcare crises [pg. 19, lines 352-371].

2. Which kind of clinical feature present in MH/SU complaints?

We appreciate the reviewers' interest in better understanding the clinical features that accompanied presenting MH/SU complaints in the present sample. While the NACRS database does collect information regarding a patient’s specific presenting complaints at the time of intake, this variable is optional in terms of NACRS data reporting guidelines and not consistently reported by participating facilities. As such, it was not possible to obtain robust data regarding specific clinical features that were present at the time of admission, so this variable was not included in the present study. Although the specific symptoms driving patients to seek emergency medical care are not known, attending physicians assign a main problem diagnosis, based on the ICD-10 classification system, to indicate the underlying condition that is most likely responsible for the presenting complaint, taking into consideration a patient’s current symptoms and medical history. Although the limitations of the current dataset prevent us from elaborating on this point, we agree that it is of great importance to further understand not only the impact of COVID-19 on specific symptoms experienced by individuals with MH/SU disorders but also which symptoms were most likely to result in the use of R/S interventions among these patients during the pandemic. To this end, we have added an additional paragraph to “Limitations” section addressing this issue [pg. 19-20, lines 370 -378].

3. What is impact of Clinical segregation of subjects on the results?

The reviewers have highlighted the importance of addressing the impact of clinical segregation on our results. When we examined the impact of COVID-19 on the use of seclusion alone, we observed a similar pattern of effects as that seen in the combined data (restraint and segregation), albeit to a lesser degree. Specifically, the probability of a seclusion event increased from pre-COVID levels by approximately 1% or less across registration month. The increase in probability for main problem diagnosis showed the same pattern as the original results in which diagnostic groups saw the greatest increases in seclusion probability, but to a smaller magnitude (~1-7% increase). These results suggest that the increases in probability seen in the combined data are primarily driven by restraint events more than seclusion events, which is consistent with the fact that there were three times more restraint events reported than seclusion events in 2019 and 2020. In the end, we chose to combine these two categories not only because they are both coercive methods of patient management and are subject to the same mandatory reporting guidelines but also because they generally show a similar pattern of effects when examined separately as they do when combined. 

4. Graphical representation is important for better representation.

We agree with the reviewer’s comment regarding the importance of graphical representation, and as such, we have added three new figures to the “Results” section to facilitate a better understanding of the data and ensure that readers can interpret the results more effectively [pg. 13-14, line 224, 227, 232; 234-248].

5. What is future prospect of the study?

To address the reviewers' comments regarding the future prospects of the study, we have expanded the discussion to include a “Future Directions” section as described above. This includes exploring additional factors associated with changes in the use of R/S interventions in response to the COVID pandemic and potential intervention strategies to prevent increased use of R/S methods during future healthcare crises [pgs. 19-21, lines 363-398].

6. Any comparative study in same region is important to understand the results.

The reviewers pointed out the importance of including comparable research conducted in the same region. To our knowledge, Martin et al. is the only other study that has looked at COVID-related changes in R/S intervention use in Ontario [described on pg. 5, lines 79-83]. The authors observed a substantial reduction in R/S intervention use, contrary to our findings in the present study. We think that the difference is because their sample was collected over a relatively short period following the onset of COVID, and as such, is not necessarily reflective of the same conditions and systemic strain as was present in Ontario EDs. 

While there does not appear to be any other comparable research on changes in R/S intervention use in EDs in the same geographical region, a number of studies from around the world have shown similar patterns of increased R/S use despite fewer people accessing care. This is consistent with the fact that COVID-19 was truly a pandemic, and like the virus itself, its impact on health care was not unique or confined to a specific geographical region. To this end, we have updated the “Introduction” section to include several newly published studies on this topic from around the world, the majority of which also found increases in R/S use [pg. 5, lines 83-88; pg. 6, lines 100-109]. 

7. What is public health model for other disease situations (epidemic-sporadic etc.)?

While we agree that it is important to consider the potential implications of our study's findings for other disease situations, the results of the present study are not as likely to be generalized to different disease scenarios, including epidemics and sporadic situations. While it is not impossible to see similar impacts on the use of R/S interventions given an epidemic of sufficient severity to overwhelm a regional health system, it is unlikely for endemic or sporadic disease events. This is because the driving factor behind the observed R/S increase during COVID was most likely the severe depletion of resources in the Canadian healthcare system, as illustrated by the fact that significant increases in R&S probability were confined to the first wave of the pandemic when the system was least prepared. Importantly, the observed increase in R&S intervention was not unique to COVID-19 or its related symptoms, but instead that its highly infectious nature and potential for severe outcomes resulted in a rapid increase in cases requiring substantial healthcare resources across the entire country (Canada) and more generally the world, in turn overwhelming an unprepared system. This was further exacerbated by the need for strict infection prevention protocols, which further strained already limited resources, both in terms of physical space and health care staff. It should be noted that public health restrictions intended to slow the spread of COVID-19 may have further exacerbated this issue by severely restricting patient access to non-emergency health care services and resources they may have previously accessed to address issues related to mental health and/or substance use concerns.

We believe that these revisions have significantly strengthened the manuscript, addressing the reviewers' concerns and improving the overall quality of the paper. We hope that these changes meet the expectations of the reviewers and the editorial board. Thank you for your time and consideration, as well as the opportunity to resubmit our revised manuscript for further evaluation.

---

## [Editor Report · Decision Letter 1]

28 Mar 2024

The impact of the COVID-19 pandemic on the use of restraint and seclusion interventions in Ontario emergency departments: A population-based study

PONE-D-23-28304R1

Dear Dr. Weissflog,

We’re pleased to inform you that your manuscript has been judged scientifically suitable for publication and will be formally accepted for publication once it meets all outstanding technical requirements.

Kind regards,

Anshuman Mishra, PhD

Academic Editor

PLOS ONE

Additional Editor Comments (optional):

Thanks for the revised version. The article is now in shape and defines more appropriately the impact of the COVID-19 pandemic on the use of restraint and seclusion interventions in Ontario.

Just one suggestion: please add the country name to the affiliation.

Decision- Accepted
---

## [Editor Report · Acceptance letter]

3 Apr 2024

PONE-D-23-28304R1 

PLOS ONE

Dear Dr. Weissflog, 

I'm pleased to inform you that your manuscript has been deemed suitable for publication in PLOS ONE. Congratulations! Your manuscript is now being handed over to our production team.

Kind regards, 

on behalf of

Dr. Anshuman Mishra 

Academic Editor

PLOS ONE